# EVALUATION-AWARE REINFORCEMENT LEARNING

## ABSTRACT

Policy evaluation is often a prerequisite for deploying safety- and performance-critical systems. Existing evaluation approaches frequently suffer from high variance due to limited data and long-horizon tasks, or high bias due to unequal support or inaccurate environmental models. We posit that these challenges arise, in part, from the standard reinforcement learning (RL) paradigm of policy learning without explicit consideration of evaluation. As an alternative, we propose evaluation-aware reinforcement learning (EvA-RL), in which a policy is trained to maximize expected return while simultaneously minimizing expected evaluation error under a given value prediction scheme—in other words, being "easy" to evaluate. We formalize a framework for EvA-RL and design an instantiation that enables accurate policy evaluation, conditioned on a small number of rollouts in an *assessment environment* that can be different than the deployment environment. However, our theoretical analysis and empirical results show that there is often a tradeoff between evaluation accuracy and policy performance when using a fixed value-prediction scheme within EvA-RL. To mitigate this tradeoff, we extend our approach to co-learn an assessment-conditioned state-value predictor alongside the policy. Empirical results across diverse discrete and continuous action domains demonstrate that EvA-RL can substantially reduce evaluation error while maintaining competitive returns. This work lays the foundation for a broad new class of RL methods that treat reliable evaluation as a first-class principle during training.

## 1 INTRODUCTION

Policy evaluation is often a prerequisite for deploying RL policies in safety- or performance-critical settings such as industrial control (Gao et al., 2014; Zheng et al., 2021), robotics (Andrychowicz et al., 2020; Kalashnikov et al., 2018) and healthcare (Komorowski et al., 2018; Fox et al., 2021). Evaluation settings are often not representative of deployment settings, due to factors such as data availability, data collection costs, and safety constraints. For example, consider a self-driving car that cannot be exhaustively tested in every possible real-world scenario that it will encounter; instead, it may be tested via a combination of limited tests in the real world (perhaps with professional drivers ready to take over), controlled scenarios such as a test course, tests in simulation, or off-policy evaluation via historical data. More generally, compared to the setting that a policy will be deployed in, evaluation data may be collected in a manner that is off-policy, limited in size, has limited support, or uses an inaccurate model of the environment.

Existing evaluation methods struggle to address these challenges: Off-policy policy evaluation (OPE), while unbiased, tends to suffer from unacceptably high variance as the horizon grows (e.g. importance sampling-based methods (Thomas et al., 2015)) or instability associated with the estimation of steady-state distributions (Liu et al., 2018). In either case, OPE assumes that the support of the behavior policy fully covers the support of the evaluation policy[1] (or similarly for the steady-state distributions) and relies on low

---

[1] A recent work by Liu & Zhang (2023) shows that transition-weighted support relaxes the requirement for full support over actions.

divergence between these quantities for efficient evaluation. Evaluation via limited on-policy data provides an unbiased estimate of performance, but is inherently high variance due to the small number of samples. Testing a fixed subset of situations that will be encountered in the deployment setting or using an inaccurate model generally leads to biased estimation.

Our core insight is that these difficulties are caused, in part, by the fact that the current dominant paradigm in AI is to first learn an arbitrary policy and then perform evaluation afterwards, without consideration for that evaluation at train-time. By contrast, **we propose evaluation-aware policy learning that is explicitly designed to learn policies that can be evaluated efficiently and accurately**. We call this evaluation-aware reinforcement learning (**EvA-RL**).

As our main contribution, we introduce a policy learning framework that steers training towards policies that maximize performance while minimizing evaluation error under a given value-prediction scheme, based on rollouts from an *assessment environment*. This assessment environment is a proxy for the deployment environment—for example, a subset of the deployment environment or a related environment such as a simulation—in which value-informative behavior can be observed more easily, cheaply, or safely than in the deployment environment. Further, we introduce a practical instantiation of this framework for accurate, low-shot policy evaluation. Specifically, an EvA-RL policy gradient learning rule is introduced that considers state-value predictions during training to learn policies with low evaluation error, while also maximizing performance.

A theoretical analysis of this framework reveals a tradeoff between maximizing policy performance and minimizing evaluation error when using a fixed value prediction scheme. We mitigate this tradeoff by co-learning a transformer-based (Vaswani et al., 2017) state-value predictor alongside the policy. This shifts part of the burden of evaluation error minimization onto the predictor itself, easing the constraints on policy learning. Our game-theoretic analysis shows convergence to optimal behavior in the limit with a near-perfect value predictor. Our empirical studies across diverse discrete and continuous control domains demonstrate that EvA-RL consistently reduces evaluation error while maintaining competitive returns. Further, we demonstrate that the co-learned predictor can be used to gate deployment to exclude high-risk policies.

Taken together, this work opens a new line of research that elevates evaluation to a first-class principle in reinforcement learning, enabling the development of policies that are both performant and supportive of efficient, accurate evaluation.

## 2 BACKGROUND AND RELATED WORK

**Markov Decision Process (MDP).** A Markov Decision Process (MDP) (Puterman, 1990) is defined by the tuple $\{\mathcal{S}, \mathcal{A}, P, R, \gamma, \mu\}$, where $\mathcal{S}$ is the state space, $\mathcal{A}$ is the action space, $P$ is the transition probability function, $R$ is the reward function, $\gamma$ is the discount factor, and $\mu$ is the start-state distribution. A policy $\pi$ specifies a distribution over actions given a state, $\pi : \mathcal{S} \to \Delta(\mathcal{A})$. We consider a finite-horizon setting, where a trajectory is $h := (s_0, a_0, r_0, \ldots, s_{T-1}, a_{T-1}, r_{T-1}, s_T)$ and its return is $g(h) := \sum_{t=0}^{T-1} \gamma^t r_t$. The state-value function of a policy $\pi$ is $V_\pi(s) := \mathbb{E}_\pi\big[\sum_{t=0}^{T-1} \gamma^t r_t \mid s_0 = s\big]$, the action-value function is $Q_\pi(s, a) := \mathbb{E}_\pi\big[\sum_{t=0}^{T-1} \gamma^t r_t \mid s_0 = s, a_0 = a\big]$, and the overall policy performance is $J_\pi := \mathbb{E}_{s\sim\mu} V_\pi(s)$.

**Policy Evaluation.** Policy evaluation refers to estimating $V_\pi(s)$ or its start-state expectation $J_\pi$. The simplest approach is Monte Carlo (MC) evaluation, where trajectories are sampled on-policy and averaged as $\hat{J}_\pi^{MC} := \frac{1}{N} \sum_{i=1}^N g(h_i)$. Since on-policy evaluation can be costly, off-policy policy evaluation (OPE) methods estimate policy performance from historical data (Uehara et al., 2022). The standard importance sampling (IS) estimator reweights trajectories from a behavior policy $\pi_b$, given by $\hat{J}_\pi^{IS} := \frac{1}{N} \sum_{i=1}^N g(h_i) \prod_{t=0}^{T^i-1} \frac{\pi(a_t^i|s_t^i)}{\pi_b(a_t^i|s_t^i)}$. Per-decision importance sampling (PDIS) (Precup et al., 2000) applies the weights step by step, leading to

$\hat{J}_\pi^{PDIS} := \frac{1}{N} \sum_{i=1}^{N} \sum_{t=0}^{T^i-1} \gamma^t r_t^i \prod_{k=0}^{t} \frac{\pi(a_k^i|s_k^i)}{\pi_b(a_k^i|s_k^i)}$, which reduces variance compared to trajectory-level IS. Fitted Q-Evaluation (FQE) (Le et al., 2019) instead learns the Q-function of the evaluation policy via Bellman consistency and derives $J_\pi$ from the learned value function. Doubly robust (DR) estimators (Jiang & Li, 2016; Thomas & Brunskill, 2016) combine PDIS with approximate value functions $\hat{Q}$ and $\hat{V}$; a common formulation is $\hat{J}_\pi^{DR} := \frac{1}{N} \sum_{i=1}^{N} \left[ \hat{V}(s_0^i) + \sum_{t=0}^{T^i-1} \gamma^t w_t^i \big( r_t^i + \gamma \hat{V}(s_{t+1}^i) - \hat{Q}(s_t^i, a_t^i) \big) \right]$, where $w_t^i = \prod_{k=0}^{t} \frac{\pi(a_k^i|s_k^i)}{\pi_b(a_k^i|s_k^i)}$. DR estimators reduce variance while maintaining consistency if either the importance weights or the value approximations are accurate. For further discussion, we refer to the survey by Uehara et al. (2022). In this work, we compare against FQE, TIS, PDIS, and DR.

A related line of work, behavior policy search (BPS) (Hanna et al., 2017; Liu & Zhang, 2022), aims to find a behavior policy that produces lower mean squared error in OPE compared to deploying the evaluation policy itself in the environment. BPS optimizes the data collection process to improve evaluation accuracy post-hoc. In contrast, our work searches for performant policies that produce lower evaluation error with respect to a co-learned value estimator. While BPS focuses on optimizing data collection for better OPE, EvA-RL integrates evaluation quality directly into policy learning, enabling policies that are inherently more evaluable.

In the upcoming section, we first introduce evaluation-aware reinforcement learning (EvA-RL). We then provide a theoretical analysis of this RL framework, followed by a practical optimization procedure for learning policies that achieve both low evaluation error and high performance.

## 3  EVALUATION-AWARE REINFORCEMENT LEARNING

The dominant approach in reinforcement learning is to learn an arbitrary policy and to consider evaluation only after learning is complete. In this section, we propose an alternative paradigm where evaluation is central to the policy learning process. Our goal is to learn policies that can be evaluated more accurately using a performance estimator, while still achieving high expected return. We refer to this paradigm as **Evaluation-Aware Reinforcement Learning (EvA-RL)**.

Consider a performance estimator $\hat{V}_\pi$ which predicts policy $\pi$'s value for state $s$ as $\hat{V}_\pi(s)$. The mean squared error (MSE): $\mathbb{E}_{s\sim\mu}\left[ \big( V_\pi(s) - \hat{V}_\pi(s) \big)^2 \right]$ forms the measure for accuracy of $\hat{V}_\pi$ predictions. With this, we formulate evaluation-aware policy learning using following objective:

$$\max_\pi \; \mathbb{E}_{s\sim\mu}\left[ V_\pi(s) - \beta \big( V_\pi(s) - \hat{V}_\pi(s) \big)^2 \right] \tag{1}$$

where $\beta > 0$ is a coefficient that trades off between maximizing expected return and minimizing evaluation error. Throughout this paper, we refer to the ability of a policy to achieve low evaluation error with respect to the estimator as its value-predictability, and to $\beta$ as the *predictability coefficient*.

In EvA-RL, we propose to estimate the policy performance using rollouts gathered in an *assessment environment*, which can be different from the deployment environment. For example, it can be a subset of the deployment environment or a related environment such as a simulation, in which value-informative behavior can be observed more easily, cheaply, or safely than in the deployment environment.

**Practical construction of assessment environments.** Many real-world control tasks already have simulation models that naturally serve as assessment environments. For instance, CARLA (Dosovitskiy et al., 2017) provides a simulator for autonomous driving, and ODE-based models (Man et al., 2014) serve as virtual patients for diabetes management. These simulators make it easy to gather informative behavior that can serve as conditioning input for predicting deployment values. Conceptually, an assessment environment can be viewed as a suite of "unit tests" for a control task, analogous to a driver's test for human drivers. A domain expert can design a set of initial states that probe behaviors of interest. In such cases, the assessment

environment shares the transition and reward dynamics with the deployment environment. It is important to note that driver's test results are reasonably predictive of broader driving performance largely due to the smooth generalization properties of humans. By contrast, an assessment of a learned neural network policy may not reflect its generalization performance in a different deployment setting. Hence, EvA-RL needs to learn a policy that behaves in a manner such that assessment behaviors are predictive of performance in the deployment environment.

Formally, the EvA-RL framework defines a deployment MDP, $\mathcal{M}_D := \{\mathcal{S}_D, \mathcal{A}_D, P_D, R_D, \gamma_D, \mu_D\}$, and an assessment MDP, $\mathcal{M}_A := \{\mathcal{S}_A, \mathcal{A}_A, P_A, R_A, \gamma_A, \mu_A\}$, such that $\mathcal{S}_D \subseteq \mathbb{R}^d$, $\mathcal{S}_A \subseteq \mathbb{R}^d$, and $\mathcal{A}_A = \mathcal{A}_D$ so that policy $\pi$ defined for $\mathcal{M}_D$ can also be executed in $\mathcal{M}_A$. We take $\mu_A$ to be a Dirac delta distribution with support only over discrete set of states, $\{s^i \mid s^i \in \mathcal{S}_A; i = 1, 2, \cdots, k\}$. This allows us to gather $k$ different assessment rollouts starting at a different state in $\{s^i\}_{i=1}^k$. The state-value function of policy $\pi$ under the deployment MDP $\mathcal{M}_D$ is denoted as $V_D^\pi$. A state-value predictor $\psi$ estimates the value of a query state $s \in \mathcal{S}_D$ using $\{h_i\}_{i=1}^k$ as $\hat{V}_D^\pi(s) = \psi(s, \{h_1, h_2, \cdots, h_k\})$. Equivalently, this mechanism can be viewed as *behavior-conditioned value prediction*, $\hat{V}_D^\pi(s) = \psi(s \mid \{h_1, h_2, \cdots, h_k\})$, where the assessment rollouts $\{h_i\}_{i=1}^k$ serve as an encoding of the policy under evaluation. This approach of conditional value prediction is conceptually similar to universal value functions (Schaul et al., 2015).

We can rewrite the EvA-RL objective as: $\quad \max_\pi \ \mathbb{E}_{s \sim \mu_D}\left[V_D^\pi(s) - \beta\left(V_D^\pi(s) - \hat{V}_D^\pi(s)\right)^2\right]$ $\quad\quad$ (2)

With $\hat{V}_D^\pi$ estimated from assessment rollouts, we use deployment interactions to approximate the target state-value function $V_D^\pi$ for policy learning. In a policy-gradient setting, for example, Monte Carlo return samples starting from state $s$ serve as target values $V_D^\pi(s)$, as in standard on-policy policy gradient algorithms such as A2C (Konda & Tsitsiklis, 1999) and TRPO (Schulman et al., 2015).

### 3.1 THEORETICAL INSIGHTS INTO EvA-RL

Now we analyze EvA-RL framework, focusing on the properties of resultant policies, their expected returns, and their evaluation errors. For our analysis, we consider $(\mathcal{S}_D, \mathcal{A}_D, P_D, R_D, \gamma_D) = (\mathcal{S}_A, \mathcal{A}_A, P_A, R_A, \gamma_A)$ with $|\mathcal{S}_D| = n$. The start distribution $\mu_A$ is assumed to have support over first $k$ states of the common state space with equal probability. Further, we consider deterministic transition and reward dynamics. The policy $\pi$ is also assumed to be deterministic. For simplicity of analysis, consider a linear transformer (Katharopoulos et al., 2020), $\psi_{\text{linear}}$, for value-prediction:

$$\psi_{\text{linear}}(s, \{h^1, h^2, \cdots, h^k\}) = \frac{\sum_{i=1}^k \phi(s)^T \phi(s^i) g(h^i)}{\sum_{i=1}^k \phi(s)^T \phi(s^i)}, \quad\quad (3)$$

where $s \in \mathcal{S}_D$, $\phi : \mathcal{S}_D \to \mathbb{R}^{d'}$ is a state-embedding function with positive inner products and $g(h^i)$ is the discounted return of assessment trajectory $h^i$. In the deterministic setting, $g(h^i) = V_D^\pi(s^i)$. While we assume deterministic dynamics for simplicity of analysis, the results naturally extend to stochastic environments and policies by conditioning the predictor on assessment state-values rather than single returns; see Appendix A.7 for details.

**Definition 1.** *The **value prediction error** of a policy $\pi$ with respect to predictor $\psi$, denoted as $\zeta_{\pi,\psi}$, is defined as the mean-squared error of its value estimates:* $\zeta_{\pi,\psi}^2 := \mathbb{E}_{s \sim \mu_D}\left[(V_D^\pi(s) - \hat{V}_D^\pi(s))^2\right]$, *where,* $\zeta_{\pi,\psi} \geq 0$ *and* $\hat{V}_D^\pi(s) = \psi(s, \{h_i \mid h_i \sim \pi; s^i \in \mathcal{S}_A; \mathcal{M}_A\}_{i=1}^k)$.

For a finite state space, we can write $\zeta_{\pi,\psi}^2 = \sum_{s \in \mathcal{S}_D} \mu_D(s)(V_D^\pi(s) - \hat{V}_D^\pi(s))^2$. We formulate a hard-constrained variant of the EvA-RL optimization objective (Eq. 2) as follows:

$$\max_\pi \ \sum_{s \in \mathcal{S}_D} \mu_D(s) V_D^\pi(s) \quad \text{s.t.} \quad \zeta_{\pi,\psi}^2 \leq \epsilon^2, \quad\quad (4)$$

where $\epsilon \geq 0$ controls the maximum permissible value prediction error. This formulation of EvA-RL allows us to characterize the overall optimization by upper bounding the prediction error. First, we characterize the hard-constrained problem by relaxing the Bellman consistency constraint on state-values and later provide insights into Bellman-consistent EvA-RL.

**Lemma 3.1.** *Bellman-relaxed, hard-constrained EvA-RL*

$$\max_{V_D^\pi(s) \in \mathbb{R}} \sum_{s \in \mathcal{S}_D} \mu_D(s) V_D^\pi(s) \quad s.t. \quad \zeta_{\pi,\psi}^2 \leq \epsilon^2 \quad \text{is a convex optimization problem.} \tag{5}$$

*Proof.* For a detailed proof, see Appendix A.1. Briefly, the optimization problem can be shown to be a quadratically constrained linear program (Boyd & Vandenberghe, 2004) with a positive semi-definite quadratic constraint. □

Thus, multiple optimal solutions for $V_D^\pi$ may exist for the Bellman-relaxed problem 5, each corresponding to a distinct policy. These solutions differ from each other by vectors in the null-space of the quadratic constraint. Incorporating Bellman-consistency constraints in the optimization may still admit multiple optimal $V_D^\pi$. This is in contrast to standard RL where the optimal value function is unique (Sutton & Barto, 1998). Additionally, the expected return under Bellman-relaxed optimization forms an upper bound on the expected return under Bellman consistency. We provide the exact expression for the upper bound in Appendix A.3.

Importantly, the Bellman-consistent hard-constrained EvA-RL optimization is not necessarily a convex optimization problem due to the possible non-convex relationship between a policy and its value (Dadashi et al., 2019).

In our work, we develop a policy gradient approach to EvA-RL optimization. The soft-constrained formulation in Eq. 2 is especially well-suited for this approach, as deployment values can be efficiently approximated using Monte Carlo samples when calculating policy gradients. To provide direct insights into our practical EvA-RL optimization, we now establish the relationship between the hard- and soft-constrained optimization problems.

**Theorem 3.1.** *Any solution $V_D^{\pi_\epsilon^*}$ to Bellman-relaxed hard-constrained problem 5 is a solution to the Bellman-relaxed soft-constrained optimization problem*

$$\max_{V_D^\pi(s) \in \mathbb{R}} \sum_{s \in \mathcal{S}_D} \mu_D(s) V_D^\pi(s) - \beta \zeta_{\pi,\psi}^2, \tag{6}$$

*for $\beta = 1/(2\epsilon^2) \times \sum_{s \in \mathcal{S}_D} \mu_D(s) V_D^{\pi_\epsilon^*}(s)$. Conversely, any solution $V_D^{\pi_\beta^*}$ to the above Bellman-relaxed soft-constrained problem 6 is also a solution to Bellman-relaxed hard-constrained problem 5 with $\epsilon = \zeta_{\pi_\beta^*,\psi}$.*

*Proof.* Refer to Appendix A.4 for a detailed proof. □

**Corollary 3.1.** *The Bellman-relaxed soft EvA-RL leads to multiple optimal solutions for $V_D^\pi$.*

*Proof.* Refer to Appendix A.4 for a detailed proof. □

Furthermore, the expected return of Bellman-relaxed soft EvA-RL can potentially be unbounded, as any solution $V_D^{\pi_\beta^*}$ can be summed with a vector from the null-space of the quadratic constraint without affecting the value prediction error. In practice, however, Bellman consistency constraints impose natural limits on the maximum achievable expected return. Analytically determining this maximum expected return remains challenging due to the inherent non-convexity of Bellman-consistent soft EvA-RL. Nevertheless, it is important to note that Bellman-consistent soft EvA-RL still permits multiple distinct value function solutions. We can further state the following:

**Proposition 3.1.** *Increasing $\beta$ in the EvA-RL optimization performed using a fixed value predictor (Eq. 2) monotonically decreases the evaluation error and the expected return of the resultant policy.*

*Proof.* Refer to Appendix A.5 for a detailed proof. $\square$

This result does not require any linearity assumption on the value-predictors and implies that *using a fixed value-predictor will lead to a trade-off between the expected return and the evaluation accuracy*.

Additionally, we provide a relationship between value prediction error $\zeta_{\pi,\psi}$ and squared error in the performance estimate of the policy $J_\pi$ using the predictor $\psi$:

**Lemma 3.2.** *The MSE in expected return of a policy is bounded above by the square of value prediction error, i.e., if $J_\pi = \mathbb{E}_{s \sim \mu_D} V_D^\pi(s)$ and $\hat{J}_\pi = \mathbb{E}_{s \sim \mu_D} \hat{V}_D^\pi(s)$, then $(J_\pi - \hat{J}_\pi)^2 \le \zeta_{\pi,\psi}^2$.*

*Proof.* Refer to Appendix A.6 for a detailed proof. $\square$

By the bound-optimization rationale, minimizing the $\zeta_{\pi,\psi}$ will reduce the provable upper bound on the error in estimation of policy performance and thereby indirectly reduce the error in the overall performance estimate.

### 3.2 PRACTICAL EvA-RL

Theory highlights that with a fixed value predictor, enforcing more accurate evaluation will inherently lower performance. To mitigate this tradeoff, we propose to co-learn the value predictor alongside the policy. Intuitively, this shifts part of the responsibility of value predictability to the predictor itself, thereby enabling less constrained policy optimization.

We introduce a general transformer-based (Vaswani et al., 2017) state-value predictor. The predictor takes as input the assessment start-states along with the corresponding policy returns and maps a query state to its value estimate in the deployment environment. In principle, the predictor can condition value predictions on entire assessment trajectories $\{h_1, h_2, \cdots, h_n\}$, but we found in practice that using only start-states and corresponding returns was sufficient. Figure 1 illustrates the EvA-RL value prediction transformer.

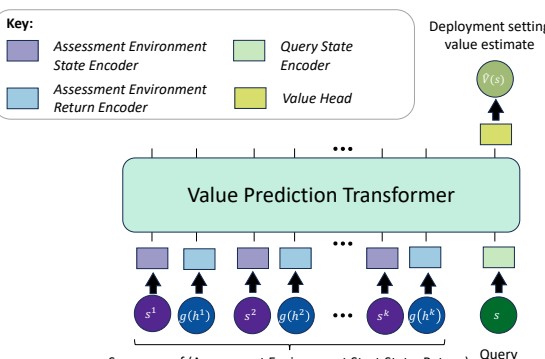

Figure 1: **Value Prediction Transformer:** A transformer encoder that takes as input (i) the start-states of the assessment environment – $\{s^1, s^2, \ldots, s^k\}$, (ii) the returns corresponding to the rollouts of policy $\pi$ starting at these states – $\{g(h_1), g(h_2), \ldots, g(h_k)\}$, and (iii) a query state $s \in \mathcal{S}_D$ and outputs an estimate for the value of the query state – $\hat{V}_D^\pi(s)$. The JAX implementation is provided in appendix B.

Let $\psi_\phi$ and $\pi_\theta$ denote the parametric value predictor and the policy, respectively. Both the policy and the predictor are initialized randomly. Hence, the initial value estimates by $\psi_\phi$ are arbitrary and cannot be used to provide any meaningful evaluation feedback. To overcome this, we train the policy for a fixed number of updates using a standard on-policy policy gradient algorithm without any evaluation feedback. From

policies $\pi_{\theta'}$ appearing during these updates, we collect assessment returns $\{g(h_A^i)|h_A^i \sim \pi_{\theta'}; s_A^i; \mathcal{M}_A\}_{i=1}^k$, deployment states $\{s_D^i|s_D^i \sim \mu_D\}_{i=1}^p$ and corresponding returns $\{g(h_D^i)|h_D^i \sim \pi_{\theta'}; s_D^i; \mathcal{M}_D\}_{i=1}^p$ in a replay buffer $\mathcal{B}_\psi$. The collected data is then used to train the value predictor, which can then be used to perform evaluation-aware learning. Further, as this learning proceeds, $\mathcal{B}_\psi$ is updated with data from the recent policies such that at any time it contains data from $m$ recent policies. We define the predictor shorthand as $\hat{V}_\phi(s_D; \Xi_A) \triangleq \psi_\phi\left(s_D, \{s_A^i\}_{i=1}^k, \{g(h_A^i)\}_{i=1}^k\right)$, where $\Xi_A = \{(s_A^i, g(h_A^i))\}_{i=1}^k$ denotes the assessment dataset. We use the following two-stage update procedure for co-learning predictor along with the policy:

$$\phi_{n+1} \leftarrow \underset{\phi}{\arg\min}\ \mathbb{E}_{(s_D, g(h_D), \Xi_A) \sim \mathcal{B}_\psi}\left[\left(g(h_D) - \hat{V}_{\phi_n}(s_D; \Xi_A)\right)^2\right],$$

$$\theta_{n+1} \leftarrow \underset{\theta}{\arg\max}\ \mathbb{E}_{s_D \sim \mu_D; h_D \sim \pi_{\theta_n}|s_D, \mathcal{M}_D; \Xi_A \sim \pi_{\theta_n}|\mathcal{M}_A}\left[g(h_D) - \beta\left(g(h_D) - \hat{V}_{\phi_{n+1}}(s_D; \Xi_A)\right)^2\right].$$

where $\alpha_\phi$ and $\alpha_\theta$ are the learning rates for the predictor and policy, respectively. For both steps, we use gradient-based updates (Eqs. 44 and 45 in appendix).

**Characterizing the updates:** The value predictor update is essentially a regression of deployment returns, conditioned on the assessment dataset $\Xi_A$. The policy update step is slightly more nuanced. We will denote the expectation $\mathbb{E}_{s_D \sim \mu_D; h_D \sim \pi_{\theta_n}|s_D, \mathcal{M}_D; \Xi_A \sim \pi_{\theta_n}|\mathcal{M}_A}$ as $\mathbb{E}_{h_D, \Xi_A \sim \pi_\theta}$. Expanding the policy update gradient yields:

$$\nabla_\theta J(\theta) = \nabla_\theta\ \mathbb{E}_{h_D, \Xi_A \sim \pi_\theta}\left[g(h_D) - \beta\left(g(h_D) - \hat{V}_\phi(s_D; \Xi_A)\right)^2\right] \tag{7}$$

$$= \underbrace{\nabla_\theta \mathbb{E}_{h_D, \Xi_A \sim \pi_\theta}\left[g(h_D)\right]}_{\text{standard policy gradient}} - 2\beta\,\mathbb{E}_{h_D, \Xi_A \sim \pi_\theta}\left[\left(g(h_D) - \hat{V}_\phi(s_D; \Xi_A)\right)\left(\nabla_\theta g(h_D) - \nabla_\theta \hat{V}_\phi(s_D; \Xi_A)\right)\right].$$

Here, $\nabla_\theta g(h_D)$ is the policy gradient computed using the deployment trajectory $h_D$ (Sutton et al., 1999). Further, $\nabla_\theta \hat{V}_\phi(s_D; \Xi_A)$ denotes the gradient of the predictor's value estimate with respect to the policy parameters and can be computed via the chain rule: $\nabla_\theta \hat{V}_\phi(s_D; \Xi_A) = \sum_{i=1}^k \frac{\partial \hat{V}_\phi(s_D; \Xi_A)}{\partial g(h_A^i)} \cdot \nabla_\theta g(h_A^i)$. This is essentially aggregation of the gradient of the value estimate with respect to each assessment return multiplied by the policy gradient computed using corresponding assessment trajectory.

The two-stage procedure leads to: (i) up-to-date value prediction by the predictor on recent policies, (ii) optimization of the policy performance in the deployment environment, (iii) preferential selection of policies that lead to lower evaluation error, and (iv) inducing assessment behavior that leads to better value predictions via the predictor. The algorithmic pseudo-code for EvA-RL is provided in appendix C.

**Convergence guarantees for co-learned predictor.** We now establish that co-learning the predictor alongside the policy yields strong convergence guarantees in the limit of extensive optimization.

**Theorem 3.2** (Convergence of Co-Learned EvA-RL). *Consider the two-stage EvA-RL optimization with co-learned predictor $\psi_\phi$ and policy $\pi_\theta$. Under standard regularity conditions (sufficient predictor capacity and adequate data coverage), the iterative updates converge to an equilibrium $(\theta^*, \phi^*)$ such that:*

*1. **Near-optimal performance:** $J(\pi_{\theta^*}) \geq J(\pi^*) - \epsilon_{perf}$, where $\pi^*$ is the optimal policy and $\epsilon_{perf} \to 0$ as the number of optimization iterations increases.*

*2. **Near-perfect evaluation:** $\zeta_{\pi_{\theta^*}, \psi_{\phi^*}} \leq \epsilon_{eval}$, where $\epsilon_{eval} \to 0$ as the number of optimization iterations increases.*

The proof, provided in Appendix A.8.1, establishes this result by modeling EvA-RL as a Stackelberg game and showing that any equilibrium of this game satisfies both properties. The key insight is that co-learning allows both $\epsilon$ values to vanish with sufficient iterations, independent of $\beta$. This theoretical guarantee justifies the empirical effectiveness of co-learning observed in our experiments.

## 4 EXPERIMENTS AND RESULTS

In this section, we present experiments designed to answer the following questions: (1) When using a fixed value predictor, does a higher predictability coefficient result in the theoretically suggested impact on returns and evaluation accuracy? And can a co-learned predictor mitigate the tradeoff between these quantities? (2) After learning a policy with EvA-RL, what is the accuracy of a co-learned value predictor compared to standard OPE? (3) How does expected return and evaluation accuracy under the full EvA-RL pipleine compare to the standard RL+OPE pipeline?

**Experimental Setup:** We evaluate EvA-RL on three discrete-action Gymnax environments (Asterix, Freeway, and Space Invaders) and three continuous-action Brax environments (HalfCheetah, Reacher, and Ant). Our implementation builds on PureJaxRL (Lu et al., 2022). Agents are trained for 10M environment interactions, and results are averaged over 5 random seeds with standard error.

**Baselines:** We choose the following OPE methods for comparison of state-value estimation: fitted Q-evaluation (FQE), trajectory importance sampling (TIS), per-decision importance sampling (PDIS), and the doubly robust (DR) estimator. For FQE and DR, we adapt code from the Scope-RL library (Kiyohara et al., 2023). We report the mean absolute error (MAE) of the value estimates using ground truth values computed via near-exhaustive on-policy evaluation.

**Assessment environment design:** We choose the assessment environment to have the same state and action spaces and same transition and reward dynamics as the deployment environment. We choose $\gamma_{ae} = 1$ and select the start states for the assessment environment by sampling 5 states from rollouts of a base RL policy trained using standard A2C. Further, we fix the horizon length for the assessment environment to be 10 steps for discrete action environments and 25 steps for continuous action environments. Example assessment start-states for MinAtar environments are given in appendix E.

Detailed list of hyperparameters used in the experiments is provided in appendix D. The code for the experiments is provided in the supplementary material.

### 4.1 RESULTS ON DISCRETE-ACTION SPACES

**(1) EvA-RL with a fixed value predictor.** We first establish that EvA-RL can learn policies with greater value predictability, and that the theoretical tradeoff between predictability and expected return manifests empirically when using a fixed value predictor. To conduct this experiment, we train a transformer-based predictor on policy data collected from all policies encountered during a standard A2C training run of 10M environment interactions. We then freeze this predictor and perform evaluation-aware policy learning from scratch. We vary the predictability coefficient $\beta$ and track both the returns and evaluation errors of the resulting policies as a function of $\beta$. The results are shown in Figure 2. We observe that the frozen predictor yields decreased evaluation error as $\beta$ increases. However, this improvement comes at the cost of reduced returns, confirming the tradeoff predicted by Corollary 3.1. Ideally, a perfect value predictor would achieve zero evaluation error and thus yield the same learning dynamics as standard RL ($\beta = 0$). However, developing such a perfect predictor is challenging in practice, and the generalization limitations of pre-training can lead to the observed tradeoff between evaluation accuracy and expected return.

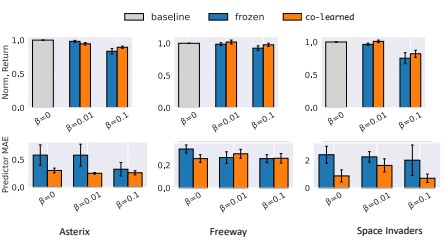

Figure 2: *EvA-RL with a pre-trained value predictor.* Top: normalized returns compared to the standard RL baseline ($\beta = 0$). Bottom: predictor MAE. With a frozen predictor, increasing $\beta$ decreases MAE but also reduces returns. Co-learning allows for better performance while maintaining low evaluation error.

Additionally, we conduct a variation of this experiment where we allow the value predictor to co-learn alongside the policy at a small learning rate. As shown in Figure 2, this approach enables us to mitigate the return-evaluation error tradeoff by improving policy performance while maintaining low evaluation error.

**(2) Comparison of co-learned value predictor with OPE methods.** The previous experiment demonstrated the benefits of co-learning the value predictor. We now investigate how well this co-learned predictor performs compared to OPE methods commonly used for policy evaluation. In this experiment, we consider a more challenging setup for EvA-RL in which no prior policy data is available to pre-train the predictor, requiring us to co-learn the value predictor alongside the policy from scratch. As described in Section 3.2, during initial iterations, we train the policy without evaluation feedback and use the collected data to train the value predictor. Once the predictor has sufficiently improved, we perform evaluation-aware policy learning with $\beta \in \{0.01, 0.1\}$. We compare the quality of value estimates from the final co-learned predictor against those from OPE methods. For fairness, we provide all OPE estimators with the same amount of data observed by the predictor during training. Figure 3 shows the comparison of evaluation errors across different performance estimators. We observe that our proposed value predictor consistently achieves much lower evaluation error compared to the OPE estimators. Furthermore, we find that as training progresses, co-learning enables the provision of high-quality value estimates during evaluation-aware learning (refer to the evaluation error learning curves in appendix F).

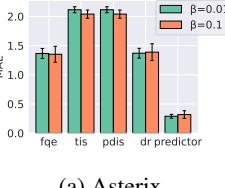
(a) Asterix

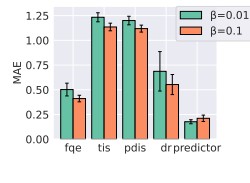
(b) Freeway

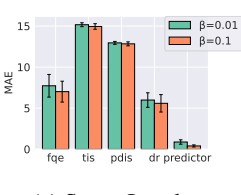
(c) Space Invaders

Figure 3: *Co-learned value predictor vs. OPE estimators for EvA-RL policies.* The predictor consistently achieves lower evaluation error showcasing the usefulness of the behavior-conditioned value prediction.

**(3) EvA-RL vs. Conventional Learning-then-Evaluation Paradigm.** Having established the utility of a co-learned value predictor in EvA-RL, we now compare the overall EvA-RL pipeline with the conventional RL approach of first learning a policy and then evaluating it (by contrast, note that in the previous experiment both evaluation methods were tested on a policy learned by the EvA-RL learning rule). For this experiment, we train one policy using the standard A2C algorithm and evaluate it with OPE methods. Concurrently, we train another policy using EvA-RL and perform evaluation using the co-learned predictor. The comparison of the resulting expected returns and evaluation errors is shown in Figure 4. We observe that EvA-RL closely matches the performance of standard RL while maintaining substantially lower evaluation error.

## 4.2 RESULTS ON CONTINUOUS-ACTION SPACES

| RL paradigm | HalfCheetah | Reacher | Ant |
|---|---|---|---|
| Standard RL w/ FQE | $29.50 \pm 9.45$ | $57.28 \pm 6.21$ | $49.56 \pm 9.77$ |
| Standard RL w/ PDIS | $\mathbf{3.31 \pm 1.61}$ | $51.33 \pm 37.37$ | $25.91 \pm 8.10$ |
| Standard RL w/ DR | $6.16 \pm 2.17$ | $32.33 \pm 5.95$ | $10.65 \pm 4.51$ |
| EvA-RL w/ Predictor | $3.46 \pm 0.94$ | $\mathbf{13.02 \pm 0.65}$ | $\mathbf{2.85 \pm 0.57}$ |

Table 1: EvA-RL often produces more accurate value estimates than OPE estimators in continuous-action environments.

| Environment | Normalized return ($\mu \pm$ SE) |
|---|---|
| HalfCheetah | $1.0361 \pm 0.0193$ |
| Reacher | $1.0001 \pm 0.0005$ |
| Ant | $0.9987 \pm 0.0035$ |

Table 2: Normalized performance of EvA-RL relative to standard A2C, with mean $\mu$ and standard error (SE) across 5 runs.

We train EvA-RL policies for continuous action tasks of half-cheetah, reacher and ant. In table 1 and table 2, we showcase results of EvA-RL with $\beta = 5 \times 10^{-4}$. Trajectory-IS estimator (TIS) is excluded as a baseline

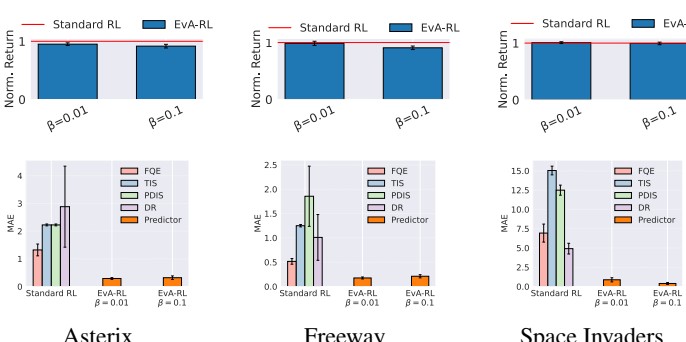

Figure 4: *Comparison of standard RL (A2C + OPE) and EvA-RL in discrete-action environments.* Top: normalized discounted returns (relative to standard RL; higher is better). Bottom: mean absolute error (MAE) of state-value estimates (lower is better). EvA-RL nearly matches standard RL in its performance while substantially reducing the evaluation error.

Asterix      Freeway      Space Invaders

due to the issue of exponential variance ($T = 1000$). We observe a repetition of the trend observed in the discrete action results – EvA-RL achieves consistent improvement in the evaluation accuracy while maintaining high returns. Learning curves for both discrete and continuous action environments are provided in appendix F.

## 4.3 A CASE STUDY ON DEPLOYMENT GATING USING EvA-RL

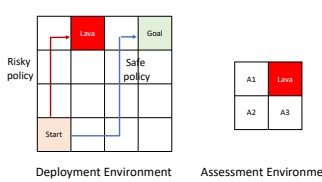

Deployment Environment     Assessment Environment

Figure 5: We consider a 4x4 deployment gridworld and a 2x2 assessment gridworld for this case study. The assessment environment has three start states A1, A2 and A3. The assessment horizon is 1. We train a linear value predictor on the assessment behavior of 3 safe policies in the deployment environment. Then, we consider two candidate policies for deployment – one safe (shown in blue) and one risky (shown in red) – and evaluate them using the trained predictor using their assessment behavior. The trained predictor's estimates (+1 for safe, -2.67 for risky) avoid deploying the risky policy without performing any additional rollouts in the deployment environment. Exact details are provided in appendix G.1.

## 5 CONCLUSION

**Implications of our results for RL development.** Our results demonstrate that evaluation-aware learning provides a compelling alternative to the conventional "learning-then-evaluation" paradigm. The reasons to use evaluation-aware learning over methods that treat evaluation post-hoc include: (i) *Data efficiency*: There are fundamental data efficiency limits to any general-purpose evaluation scheme that works for arbitrary policies. By considering evaluation at learning-time and co-learning a policy/predictor pair, we can surpass these limits. This is a unique feature of our method, and even though theoretical speculations exist in the literature (Brown et al., 2021), our work is the first empirical instantiation of using assessment tests (analogous to driver's tests) for general control tasks. (ii) *Empirical effectiveness*: EvA-RL significantly reduces the mean-squared error in policy evaluation compared to OPE methods. (iii) *Safety-critical applications*: As demonstrated in our gridworld case study (Appendix G.1), assessment behavior can gate deployment to exclude high-risk policies. (iv) *Theoretical guarantees*: Our game-theoretic analysis (Appendix A.8) shows convergence to optimal behavior in the limit with a near-perfect value predictor.

**Limitations.** Our work depended on randomly sampled states from base policy for defining the assessment start-states. While effective in our experiments, systematic design may lead to stronger behavioral encodings and improved evaluation. In addition, our predictor used only states and returns from the assessment rollouts, rather than full trajectories. Although this design proved sufficient, richer inputs could yield more accurate predictors.

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

## A   THEOREMS AND PROOFS

This appendix provides a detailed theoretical analysis of the predictability objectives used in EvA-RL, including vectorized formulations, convexity proofs, monotonicity properties, and the relationship between soft and hard constraints.

### A.1   VECTORIZED FORMULATION OF EvA-RL OBJECTIVES FOR ANALYSIS

To facilitate theoretical analysis, we first recast the EvA-RL objective in a vectorized form. This approach enables us to leverage linear algebraic tools for understanding the structure and properties of the optimization problem.

Recall the soft EvA-RL objective:

$$\max_{\pi} \ \mathbb{E}_{s\sim\mu_D}\left[V_D^\pi(s) - \beta\big(V_D^\pi(s) - \hat{V}_D^\pi(s)\big)^2\right]. \tag{8}$$

Let $\mu_D = [\mu_D(s_1), \mu_D(s_2), \ldots, \mu_D(s_n)]^T$ denote the start-state distribution, and $V_D^\pi = [V_D^\pi(s_1), \ldots, V_D^\pi(s_n)]^T$ the vector of state values. Without loss of generality, we assume the first $k$ states correspond to the start-states of the assessment environment $\mathcal{S}_A$.

We use a pre-trained linear transformer $\psi_{\text{linear}}$ (see Eq. 3 in the main text), and define a similarity function $f : \mathcal{S}_D \times \mathcal{S}_D \to \mathbb{R}^+$ as $f(s, s') = \phi(s)^T\phi(s')$. The transformer can then be written as:

$$\hat{V}_D^\pi(s) = \psi_{\text{linear}}(s, \{h_1, \ldots, h_k\}) = \frac{\sum_{s^i \in \mathcal{S}_A} f(s, s^i)\, g(h^i)}{\sum_{s^i \in \mathcal{S}_A} f(s, s^i)}. \tag{9}$$

To express this in matrix form, we construct a matrix $F$ of dimensions $n \times n$ as follows:

$$F = \begin{bmatrix} f(s^1, s^1) & f(s^1, s^2) & \cdots & f(s^1, s^k) & 0 & 0 & \cdots & 0 \\ f(s^2, s^1) & f(s^2, s^2) & \cdots & f(s^2, s^k) & 0 & 0 & \cdots & 0 \\ \vdots & \vdots & \ddots & \vdots & \vdots & \vdots & \ddots & \vdots \\ f(s^k, s^1) & f(s^k, s^2) & \cdots & f(s^k, s^k) & 0 & 0 & \cdots & 0 \\ f(s_{k+1}, s^1) & f(s_{k+1}, s^2) & \cdots & f(s_{k+1}, s^k) & 0 & 0 & \cdots & 0 \\ f(s_{k+2}, s^1) & f(s_{k+2}, s^2) & \cdots & f(s_{k+2}, s^k) & 0 & 0 & \cdots & 0 \\ \vdots & \vdots & \ddots & \vdots & \vdots & \vdots & \ddots & \vdots \\ f(s_n, s^1) & f(s_n, s^2) & \cdots & f(s_n, s^k) & 0 & 0 & \cdots & 0 \end{bmatrix}_{n \times n} \tag{10}$$

We use superscripted $s$ to denote the start states of the deployment/assessment environment, for the rest of the states, we use subscripted $s$.

This allows us to write the vector of EvA-RL value estimates as

$$\hat{V}_D^\pi = \text{diag}(F1_n)^{-1} F V_D^\pi, \tag{11}$$

where $1_n$ is a vector of ones of size $n$, and $\text{diag}(F1_n)$ is a diagonal matrix with $F1_n$ on its diagonal.

The expected mean-squared error can then be written as

$$\mathbb{E}_{s\sim\mu_D}[V_D^\pi(s) - \hat{V}_D^\pi(s)]^2 = V_D^{\pi T}(I - \text{diag}(F1_n)^{-1}F)^T \text{diag}(\mu_D)(I - \text{diag}(F1_n)^{-1}F)V_D^\pi, \tag{12}$$

where $\text{diag}(\mu_D)$ is a diagonal matrix with $\mu_D$ on its diagonal.

With this vectorized representation, we can now analyze the properties of the resulting optimization problem, as detailed in the following sections.

We can express the soft EvA-RL objective in equation 8 as:

$$\max_{\pi} \quad \mu_D^T V_D^\pi - \beta[V_D^{\pi T}(I - \text{diag}(F1_n)^{-1}F)^T \text{diag}(\mu_D)(I - \text{diag}(F1_n)^{-1}F)V_D^\pi] \tag{13}$$

The hard-constrained EvA-RL objective can be written as:

$$\max_{\pi} \quad \mu_D^T V_D^\pi \quad \text{s.t.} \quad V_D^{\pi T}(I - \text{diag}(F1_n)^{-1}F)^T \text{diag}(\mu_D)(I - \text{diag}(F1_n)^{-1}F)V_D^\pi \leq \epsilon^2. \tag{14}$$

## A.2 Analysis of the optimization problem

**Lemma A.1.**

$$\max_{V_D^\pi(s) \in \mathbb{R}} \sum_{s \in \mathcal{S}_D} \mu_D(s)V_D^\pi(s) \text{ s.t. } \sum_{s \in \mathcal{S}_D} \mu_D(s)(V_D^\pi(s) - \hat{V}_D^\pi(s))^2 \leq \epsilon^2 \text{ is a convex optimization problem.}$$

$$\tag{15}$$

*Proof.* As described in the earlier section of the appendix A.1, the vectorized form of given optimization problem can be written as:

$$\max_{V_D^\pi \in \mathbb{R}^n} \quad \mu_D^T V_D^\pi \quad \text{s.t.} \quad V_D^{\pi T}(I - \text{diag}(F1_n)^{-1}F)^T \text{diag}(\mu_D)(I - \text{diag}(F1_n)^{-1}F)V_D^\pi \leq \epsilon^2 \tag{16}$$

In this optimization problem, we have the objective to maximize $\mu_D^T V_D^\pi$, an affine function of $V_D^\pi$.

Now we will characterise the predictability constraint. We can observe that this constraint is quadratic in nature. Further, we have $\forall V_D^\pi$,

$$V_D^{\pi T}(I - \text{diag}(F1_n)^{-1}F)^T \text{diag}(\mu_D)(I - \text{diag}(F1_n)^{-1}F)V_D^\pi \geq 0.$$

This means that the matrix $(I - \text{diag}(F1_n)^{-1}F)^T \text{diag}(\mu_D)(I - \text{diag}(F1_n)^{-1}F)$ is a positive semi-definite matrix. As a consequence, the quadratic constraint is also a convex constraint.

Therefore, the problem is an instance of convex quadratic program (QP) over state-values (Boyd & Vandenberghe, 2004). It can further be categorized as a quadratically constrained linear program (QCLP). $\square$

## A.3 Upper bound on the expected return in Bellman-consistent EvA-RL

The value of the objective function at the solution for a problem of the form:

$$\max_{x} \quad a^T x \quad \text{s.t.} \quad x^T Q x \leq \epsilon^2 \tag{17}$$

where $a \in \mathbb{R}^n$ and $Q \in \mathbb{R}^{n \times n}$ is a positive semi-definite matrix, is given by:

$$a^T x^* = \epsilon \sqrt{a^T Q^\dagger a} \tag{18}$$

where $Q^\dagger$ is the pseudo-inverse of $Q$.

We know that the expected return under Bellman consistency is strictly upper bounded by the expected return under Bellman-relaxed optimization. This bound is given by:

$$\epsilon \sqrt{\mu_D^T[(I - \text{diag}(F1_n)^{-1}F)^T \text{diag}(\mu_D)(I - \text{diag}(F1_n)^{-1}F)]^\dagger \mu_D}$$

.

## A.4 RELATIONSHIP BETWEEN SOFT AND HARD CONSTRAINED PREDICTABILITY OPTIMIZATION

**Theorem A.1.** *Any solution $V_D^{\pi_\epsilon^*}$ to Bellman-relaxed hard-constrained problem 5 is a solution to the Bellman-relaxed soft-constrained optimization problem*

$$\max_{V_D^\pi(s) \in \mathbb{R}} \quad \sum_{s \in \mathcal{S}_D} \mu_D(s) V_D^\pi(s) - \beta \zeta_{\pi,\psi}^2, \tag{19}$$

*for $\beta = 1/(2\epsilon^2) \times \sum_{s \in \mathcal{S}_D} \mu_D(s) V_D^{\pi_\epsilon^*}(s)$. Conversely, any solution $V_D^{\pi_\beta^*}$ to the above Bellman-relaxed soft-constrained problem is also a solution to Bellman-relaxed hard-constrained problem 5 with $\epsilon = \zeta_{\pi_\beta^*,\psi}$.*

*Proof.* We can write the the objective $\max_{V_D^\pi} \sum_{s \in \mathcal{S}_D} \mu_D(s) V_D^\pi(s) - \beta \sum_{s \in \mathcal{S}_D} \mu_D(s)(V_D^\pi(s) - \hat{V}_D^\pi(s))^2$ in a vectorized form as,

$$\max_{V_D^\pi} \quad \mu_D^T V_D^\pi - \beta V_D^{\pi T}(I - \text{diag}(F 1_n)^{-1} F)^T \text{diag}(\mu_D)(I - \text{diag}(F 1_n)^{-1} F) V_D^\pi \tag{20}$$

Let $(I - \text{diag}(F 1_n)^{-1} F)^T \text{diag}(\mu_D)(I - \text{diag}(F 1_n)^{-1} F) = Q$. We have $Q$ to be positive semi-definite matrix from earlier proof A.1. Then, the optimization problem can be rewritten as:

$$\max_{V_D^\pi} \quad \mu_D^T V_D^\pi - \beta V_D^{\pi T} Q V_D^\pi \tag{21}$$

This is a quadratic optimization problem with positive semidefinite matrix corresponding to the quadratic term. The solutions of this optimization problem can be characterized by the first-order optimality condition:

$$\nabla_{V_D^\pi} \left( \mu_D^T V_D^\pi - \beta V_D^{\pi T} Q V_D^\pi \right) = 0 \implies \mu_D - 2\beta Q V_D^\pi = 0 \implies Q V_D^\pi = \frac{1}{2\beta} \mu_D \tag{22}$$

We have $V_D^{\pi*_\beta}$ as one of the solutions to the above equation implying $Q V_D^{\pi_\beta^*} = \frac{1}{2\beta} \mu_D$.

We set

$$\epsilon^2 = \sum_{s \in \mathcal{S}_D} \mu_D(s)(V_D^{\pi_\beta^*}(s) - \hat{V}_D^{\pi_\beta^*}(s))^2 \tag{23}$$

$$= V_D^{\pi_\beta^* T}(I - \text{diag}(F 1_n)^{-1} F)^T \text{diag}(\mu_D)(I - \text{diag}(F 1_n)^{-1} F) V_D^{\pi_\beta^*} \tag{24}$$

To prove that for this $\epsilon$, $V_D^{\pi_\beta^*}$ is also a solution to the hard-constrained optimization problem, consider a global optima of hard-constrained optimization problem $V_D^{\pi_\epsilon^*}$. If $V_D^{\pi_\epsilon^*} \neq V_D^{\pi_\beta^*}$, then

$$\mu_D^T V_D^{\pi_\epsilon^*} - \beta V_D^{\pi_\epsilon^* T} Q V_D^{\pi_\epsilon^*} > \mu_D^T V_D^{\pi_\beta^*} - \beta V_D^{\pi_\beta^* T} Q V_D^{\pi_\beta^*}.$$

This implies that $V_D^{\pi_\beta^*}$ is not a maximizer of the soft-constrained optimization problem – a contradiction. Hence, $V_D^{\pi_\beta^*}$ is also a solution to the hard-constrained optimization problem.

Also, consider Lagrangian of the hard-constrained optimization problem:

$$\mathcal{L}(V_D^\pi, \lambda) = -\mu_D^T V_D^\pi + \lambda \left( V_D^{\pi T} Q V_D^\pi - \epsilon^2 \right) \tag{25}$$

At the optimum, we have:

$$\nabla_{V_D^\pi} \mathcal{L}(V_D^{\pi_\beta^*}, \lambda) = 0 \implies -\mu_D + 2\lambda Q V_D^{\pi_\beta^*} = 0 \implies Q V_D^{\pi_\beta^*} = \frac{1}{2\lambda} \mu_D \tag{26}$$

Comparing this with $QV_D^{\pi_\beta^*} = \frac{1}{2\beta}\mu_D$, we have $\lambda = \beta$.

Moreover, we have $\epsilon^2 = V_D^{\pi_\beta^* T} QV_D^{\pi_\beta^*} = V_D^{\pi_\beta^* T} \mu_D/(2\beta)$, which implies $\beta = \mu_D^T V_D^{\pi_\beta^*}/(2\epsilon^2)$. $\qquad\square$

**Corollary A.1.** *The Bellman-relaxed soft EvA-RL leads to multiple optimal solutions for $V_D^\pi$.*

*Proof.* As the proof of above theorem suggests, any $V_D^\pi \in \mathbb{R}^n$ that satisfies $QV_D^\pi = \frac{1}{2\beta}\mu_D$ is a valid solution to the optimization problem. Since $Q$ is positive semi-definite, the number of $V_D^\pi$'s that satisfy the above equation is infinite. $\qquad\square$

### A.5 MONOTONICITY OF PREDICTABILITY GAP AND RETURNS IN PREDICTABILITY OPTIMIZATIONS

**Lemma A.2.** *Let $x^*(\beta) \in \arg\max_x \{f(x) - \beta\, g(x)\}$ with $\beta \geq 0$. Then $g(x^*(\beta))$ is non-increasing in $\beta$; i.e. for $\beta_2 > \beta_1 \geq 0$, $g(x^*(\beta_2)) \leq g(x^*(\beta_1))$.*

*Proof.* Set $x_1 = x^*(\beta_1)$, $x_2 = x^*(\beta_2)$ with $\beta_2 > \beta_1$. Optimality gives

$$f(x_1) - \beta_1 g(x_1) \geq f(x_2) - \beta_1 g(x_2), \qquad f(x_2) - \beta_2 g(x_2) \geq f(x_1) - \beta_2 g(x_1).$$

If $g(x_2) > g(x_1)$, adding the two inequalities yields $0 \geq (\beta_2 - \beta_1)(g(x_2) - g(x_1)) > 0$, a contradiction; hence $g(x_2) \leq g(x_1)$. $\qquad\square$

**Lemma A.3.** *Under the same setup, $f(x^*(\beta))$ is non-increasing in $\beta$; explicitly, for $\beta_2 > \beta_1 \geq 0$, $f(x^*(\beta_2)) \leq f(x^*(\beta_1))$.*

*Proof.* With $x_1, x_2$ as above, Lemma A.2 gives $g(x_1) \geq g(x_2)$. Optimality at $\beta_1$ implies

$$f(x_1) - \beta_1 g(x_1) \geq f(x_2) - \beta_1 g(x_2) \implies f(x_1) - f(x_2) \geq \beta_1(g(x_1) - g(x_2)) \geq 0.$$

Hence $f(x_2) \leq f(x_1)$. $\qquad\square$

**Proposition A.1.** *Increasing $\beta$ in the EvA-RL optimization performed using a fixed value predictor (Eq. 2) monotonically decreases the evaluation error and the expected return of the resultant policy.*

*Proof.* Consider EvA-RL predictability objective 2 written below:

$$\max_\pi \sum_{s \in \mathcal{S}_D} \mu_D(s) V_D^\pi(s) - \beta \sum_{s \in \mathcal{S}_D} \mu_D(s)(V_D^\pi(s) - \hat{V}_D^\pi(s))^2 \tag{27}$$

and its simplified version 6 where we search for real-valued value functions:

$$\max_{V_D^\pi \in \mathbb{R}^n} \sum_{s \in \mathcal{S}_D} \mu_D(s) V_D^\pi(s) - \beta \sum_{s \in \mathcal{S}_D} \mu_D(s)(V_D^\pi(s) - \hat{V}_D^\pi(s))^2 \tag{28}$$

Let $0 \leq \beta_1 < \beta_2$. If $V_D^{\pi_{\beta_1}^*}$ and $V_D^{\pi_{\beta_2}^*}$ are optima of the simplified soft predictability objective, we have from lemmas A.2 and A.3, $\mu_D^T V_D^{\pi_{\beta_1}^*} \leq \mu_D^T V_D^{\pi_{\beta_2}^*}$. Also, if $\zeta_{\pi_{\beta_1}^*, \psi}$ and $\zeta_{\pi_{\beta_2}^*, \psi}$ are the evaluation errors corresponding to $V_D^{\pi_{\beta_1}^*}$ and $V_D^{\pi_{\beta_2}^*}$, we have $\zeta_{\pi_{\beta_1}^*, \psi} < \zeta_{\pi_{\beta_2}^*, \psi}$.

When we consider additional Bellman constraints for characterizing the return and the evaluation error of the original soft predictability problem over policies, the monotonicity results on simplified optimization objective straightforwardly extend as lemmas A.2 and A.3 work with any feasible set of $x$, possibly defined by constraints. $\qquad\square$

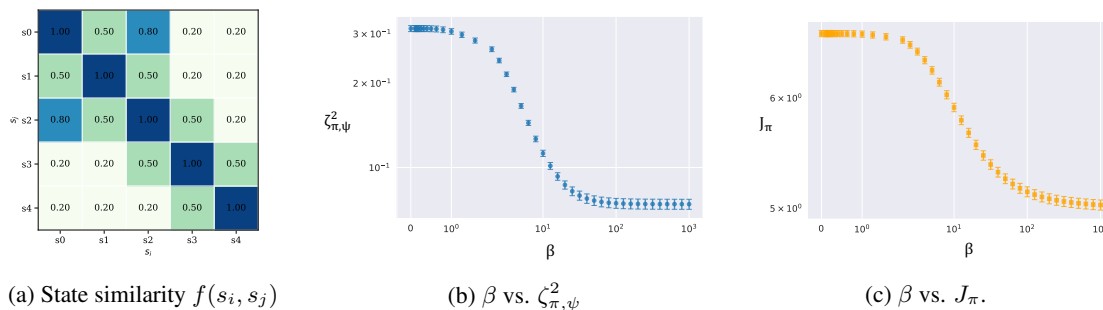

(a) State similarity $f(s_i, s_j)$     (b) $\beta$ vs. $\zeta_{\pi,\psi}^2$     (c) $\beta$ vs. $J_\pi$.

Figure 6: **Effect of predictability coefficient $\beta$ on the resultant evaluation error, $\zeta_{\pi,\psi}$ and expected return, $J_\pi$.** We design a custom MDP with 5 states and 2 actions per state. The state similarity function for this MDP is fixed as shown in sub-figure 6a. States $s_0$ and $s_2$ together form the assessment environment. We use uniform distribution over states as our start-state distribution and set the discount factor to 0.9. For each trial, we randomly sample the transition and reward dynamics of this MDP, and solve soft predictability optimization by training a softmax policy for a chosen value of $\beta$. In subfigures 6b and 6c, we report the resultant squared evaluation error $\zeta_{\pi,\psi}^2$ and the expected return $J_\pi$, respectively. Each data point depicts the average of 1000 random trials along with standard error for each value of $\beta$. We can observer that both the gap and the return monotonically decrease as $\beta$ increases.

## A.6    POLICY EVALUATION OF EVA-RL POLICIES

**Lemma A.4.** *The squared error in estimation of the true performance of the policy: $J_\pi = \mathbb{E}_{s \sim \mu_D} V_D^\pi(s)$ using predictor's value estimates: $\hat{J}_\pi = \mathbb{E}_{s \sim \mu_D} \hat{V}_D^\pi(s)$, is bounded above by the squared-value prediction error: $(J_\pi - \hat{J}_\pi)^2 \leq \zeta_{\pi,\psi}^2$.*

*Proof.*

$$
\begin{aligned}
(J_\pi - \hat{J}_\pi)^2 &= \Big( \sum_{s \in \mathcal{S}_D} \mu_D(s) V_D^\pi(s) - \sum_{s \in \mathcal{S}_D} \mu_D(s) \hat{V}_D^\pi(s) \Big)^2 = \Big( \sum_{s \in \mathcal{S}_D} \mu_D(s)(V_D^\pi(s) - \hat{V}_D^\pi(s)) \Big)^2 \\
&= \sum_{s_1, s_2 \in \mathcal{S}_D} \mu_D(s_1)\mu_D(s_2)(V_D^\pi(s_1) - \hat{V}_D^\pi(s_1))(V_D^\pi(s_2) - \hat{V}_D^\pi(s_2)) \\
&\leq \sum_{s_1, s_2 \in \mathcal{S}_D} \mu_D(s_1)\mu_D(s_2)|V_D^\pi(s_1) - \hat{V}_D^\pi(s_1)||V_D^\pi(s_2) - \hat{V}_D^\pi(s_2)| \\
&\leq \sum_{s_1, s_2 \in \mathcal{S}_D} \mu_D(s_1)\mu_D(s_2)((V_D^\pi(s_1) - \hat{V}_D^\pi(s_1))^2 + (V_D^\pi(s_2) - \hat{V}_D^\pi(s_2))^2)/2 \\
&= \sum_{s \in \mathcal{S}_D} \mu_D(s)(V_D^\pi(s) - \hat{V}_D^\pi(s))^2 = \zeta_{\pi,\psi}^2
\end{aligned}
\tag{29}
$$

Here for the first inequality we used the Cauchy-Schwarz inequality and for the second we used AM-GM inequality. The above result can also be written as: $J_\pi \in [\hat{J}_\pi - \zeta_{\pi,\psi}, \hat{J}_\pi + \zeta_{\pi,\psi}]$. $\qquad\square$

## A.7 EXTENSION TO STOCHASTIC ENVIRONMENTS AND POLICIES

The theoretical analysis presented in the main text assumes deterministic transition and reward dynamics, as well as deterministic policies. We now show how this analysis naturally extends to stochastic environments and policies.

**Generalized predictor for stochastic settings.** The key insight is to generalize the linear predictor to condition on assessment state-values rather than single-sample returns. Specifically, we can write:

$$\psi_{\text{linear}}(s \mid V_A^\pi(s^1), V_A^\pi(s^2), \ldots, V_A^\pi(s^k)) = \frac{\sum_{i=1}^k \phi(s)^T \phi(s^i) V_A^\pi(s^i)}{\sum_{i=1}^k \phi(s)^T \phi(s^i)}, \tag{30}$$

where $V_A^\pi(s^i)$ is the true value of the assessment state $s^i$ under policy $\pi$ in the assessment MDP $\mathcal{M}_A$. This formulation replaces the single Monte Carlo return $g(h^i)$ used in the deterministic case with the expected value $V_A^\pi(s^i) = \mathbb{E}_{\pi, \mathcal{M}_A}[g(h) \mid s_0 = s^i]$.

**Practical approximation.** By design, assessment rollouts are computationally inexpensive, so we can draw many assessment trajectories starting from each state $s^i$ to closely approximate these values at low cost:

$$V_A^\pi(s^i) \approx \frac{1}{M} \sum_{j=1}^M g(h^{i,j}), \tag{31}$$

where $h^{i,j}$ denotes the $j$-th trajectory sampled from state $s^i$ under policy $\pi$ in $\mathcal{M}_A$. Using values in the predictor remains consistent with behavior-conditioned prediction, since values summarize the quality of expected behavior under the policy.

**Empirical implementation.** Empirically, feeding assessment returns into the transformer (as described in Section 3.2) can be viewed as a single-sample approximation of these values, i.e., $M = 1$. While this is an approximation, it proves effective in practice. For environments with high stochasticity, one can increase $M$ to obtain more accurate value estimates at the cost of additional assessment rollouts.

**Theoretical properties preserved.** The earlier determinism assumption simply allowed us to treat a single Monte Carlo return from an assessment state as its exact value. As long as we have access to sufficiently accurate approximations of assessment state-values (either through multiple samples or low environment stochasticity), the key theoretical results established in our analysis continue to hold:

- **Convexity (Lemma 3.1):** The Bellman-relaxed hard-constrained optimization remains a convex quadratically constrained linear program when values are used instead of single returns.

- **Monotonicity (Proposition 3.1):** The monotonic relationship between $\beta$ and both evaluation error and expected return continues to hold, as the predictor structure remains unchanged.

- **Performance bounds (Lemma 3.2):** The bound $(J_\pi - \hat{J}_\pi)^2 \leq \zeta_{\pi,\psi}^2$ remains valid, as it depends only on the predictor's mean squared error, not on the determinism of the environment.

**Remark on idealization.** While exact values $V_A^\pi(s^i)$ may still be hard to compute in some highly stochastic environments, this is a standard theoretical simplification whose insights carry over to more realistic settings. The practical EvA-RL algorithm (Algorithm 1 in Appendix C) naturally handles stochasticity by using Monte Carlo samples, and our empirical results on stochastic environments (Section 4) demonstrate the effectiveness of this approach.

## A.8 GAME-THEORETIC ANALYSIS OF EvA-RL WITH CO-LEARNED PREDICTOR

We now provide a game-theoretic characterization of EvA-RL when the predictor is co-learned alongside the policy. This analysis addresses the question of what happens under extensive exploration and optimization iterations.

**Stackelberg game formulation.** We model EvA-RL with a co-learned predictor as a Stackelberg game (Rajeswaran et al., 2020; Zheng et al., 2022; Zhang et al., 2020) between two players: the policy (leader) and the predictor (follower). Stackelberg games are sequential leader–follower games and have been used to model various RL frameworks, including model-based RL, policy optimization, and multi-agent settings.

In our setting, the policy updates to maximize return while keeping evaluation error low with respect to the predictor, and the predictor updates to reduce evaluation error given the policy's assessment behavior. Algorithm 1 in Appendix C can thus be seen as a "policy-as-leader" Stackelberg game with the following structure:

- **Leader (Policy):** The policy $\pi_\theta$ chooses actions to maximize

$$\max_\theta \ \mathbb{E}_{s \sim \mu_D}\left[ V_D^{\pi_\theta}(s) - \beta\left(V_D^{\pi_\theta}(s) - \hat{V}_\phi^{\pi_\theta}(s)\right)^2 \right], \tag{32}$$

  where the predictor $\psi_\phi$ is treated as given (but adapting).

- **Follower (Predictor):** The predictor $\psi_\phi$ minimizes prediction error on the current policy:

$$\min_\phi \ \mathbb{E}_{(s_D, g(h_D), \Xi_A) \sim \mathcal{B}_\psi}\left[ \left(g(h_D) - \hat{V}_\phi(s_D; \Xi_A)\right)^2 \right]. \tag{33}$$

**Nash equilibrium characterization.** The predictor is continuously improving to better evaluate recent policies, while the policy improves its expected return within the implicit "trust region" defined by predictor accuracy. Neither player has an incentive to act adversarially: the policy gains by improving return under the given evaluation penalty, and the predictor gains by reducing error on the evolving policy distribution stored in $\mathcal{B}_\psi$.

In the limit of many optimization iterations with sufficient exploration, the policy converges to a return-maximizing solution, and the predictor accurately evaluates this final policy. At this point, neither player benefits from deviating, yielding a Nash equilibrium for the Stackelberg game. Formally, at equilibrium $(\theta^*, \phi^*)$:

- The policy $\pi_{\theta^*}$ achieves near-optimal expected return: $J(\pi_{\theta^*}) \approx \max_\pi J(\pi)$, subject to the evaluation constraint.
- The predictor $\psi_{\phi^*}$ achieves near-zero evaluation error: $\zeta_{\pi_{\theta^*}, \psi_{\phi^*}} \approx 0$.

Under well-behaved predictor co-learning (e.g., sufficient capacity, appropriate learning rates, and adequate data coverage in $\mathcal{B}_\psi$), this equilibrium is unique. The policy cannot improve return without increasing evaluation error beyond what the predictor can accommodate, and the predictor cannot reduce error further on the equilibrium policy. This game-theoretic view provides a principled characterization of the long-term behavior of EvA-RL and explains why co-learning the predictor enables both high performance and low evaluation error.

### A.8.1 FORMAL CONVERGENCE RESULT

We now provide a formal proof of Theorem 3.2, which establishes convergence guarantees for co-learned EvA-RL.

**Theorem A.2** (Convergence of Co-Learned EvA-RL (Restated)). *Consider the two-stage EvA-RL optimization with co-learned predictor $\psi_\phi$ and policy $\pi_\theta$. Under the following regularity conditions:*

- *(C1) Sufficient predictor capacity: The predictor class $\{\psi_\phi : \phi \in \Phi\}$ is rich enough to approximate any continuous function on the state space to arbitrary accuracy.*

- *(C2) Sufficient data coverage: The replay buffer $\mathcal{B}_\psi$ maintains data from a sliding window of recent policies with sufficient diversity.*

*Then, the iterative updates converge to an equilibrium $(\theta^*, \phi^*)$ such that:*

1. *Near-optimal performance: $J(\pi_{\theta^*}) \geq J(\pi^*) - \epsilon_{perf}$, where $\pi^*$ is the optimal policy and $\epsilon_{perf} \to 0$ as the number of optimization iterations $T \to \infty$.*

2. *Near-perfect evaluation: $\zeta_{\pi_{\theta^*}, \psi_{\phi^*}} \leq \epsilon_{eval}$, where $\epsilon_{eval} \to 0$ as the number of optimization iterations $T \to \infty$.*

*Proof.* We establish convergence by analyzing the Stackelberg game structure and showing that any equilibrium satisfies both properties.

**Step 1: Predictor convergence (Follower's best response).**

Given a fixed policy $\pi_\theta$, the predictor update minimizes:

$$L(\phi) = \mathbb{E}_{(s_D, g(h_D), \Xi_A) \sim \mathcal{B}_\psi} \left[ \left( g(h_D) - \hat{V}_\phi(s_D; \Xi_A) \right)^2 \right]. \tag{34}$$

Under condition (C1), for any fixed policy $\pi_\theta$, there exists $\phi^*(\theta)$ such that:

$$\lim_{t \to \infty} \mathbb{E}_{(s_D, g(h_D), \Xi_A) \sim \pi_\theta} \left[ \left( V_D^{\pi_\theta}(s_D) - \hat{V}_{\phi^*(\theta)}(s_D; \Xi_A) \right)^2 \right] = 0. \tag{35}$$

Under condition (C2), the predictor parameters converge: $\phi_t \to \phi^*(\theta_t)$ as $t \to \infty$.

**Step 2: Policy convergence (Leader's best response).**

The policy update maximizes:

$$J(\theta) = \mathbb{E}_{s_D \sim \mu_D; h_D \sim \pi_\theta} \left[ g(h_D) - \beta \left( g(h_D) - \hat{V}_{\phi^*(\theta)}(s_D; \Xi_A) \right)^2 \right]. \tag{36}$$

We decompose this objective as:

$$J(\theta) = \mathbb{E}_{s_D \sim \mu_D} \left[ V_D^{\pi_\theta}(s_D) \right] - \beta \mathbb{E}_{s_D \sim \mu_D} \left[ \left( V_D^{\pi_\theta}(s_D) - \hat{V}_{\phi^*(\theta)}(s_D; \Xi_A) \right)^2 \right] \tag{37}$$

$$= J(\pi_\theta) - \beta \zeta_{\pi_\theta, \psi_{\phi^*(\theta)}}^2. \tag{38}$$

At equilibrium $\theta^*$, the policy satisfies the first-order optimality condition:

$$\nabla_\theta J(\theta^*) = \nabla_\theta J(\pi_{\theta^*}) - 2\beta \nabla_\theta \zeta_{\pi_{\theta^*}, \psi_{\phi^*(\theta^*)}}^2 = 0. \tag{39}$$

**Step 3: Equilibrium characterization.**

At any equilibrium $(\theta^*, \phi^*)$, neither player can improve unilaterally:

- The predictor $\phi^*$ minimizes prediction error on policy $\pi_{\theta^*}$: $\phi^* = \phi^*(\theta^*)$.

- The policy $\theta^*$ maximizes $J(\theta) - \beta\zeta^2_{\pi_\theta,\psi_{\phi^*}}$ given predictor $\phi^*$.

**Step 4: Near-perfect evaluation.**

From Step 1, as $t \to \infty$, the predictor converges to minimize prediction error on the current policy. At any equilibrium:

$$\zeta^2_{\pi_{\theta^*},\psi_{\phi^*}} = \inf_{\phi\in\Phi} \mathbb{E}_{s_D\sim\mu_D}\left[\left(V_D^{\pi_{\theta^*}}(s_D) - \hat{V}_\phi(s_D;\Xi_A)\right)^2\right]. \tag{40}$$

Under condition (C1), this infimum can be made arbitrarily small, hence $\zeta_{\pi_{\theta^*},\psi_{\phi^*}} \le \epsilon_{\text{eval}}$ where $\epsilon_{\text{eval}} \to 0$ as predictor capacity and training iterations increase.

**Step 5: Near-optimal performance.**

Consider the optimal policy $\pi^*$ that maximizes $J(\pi)$ without the evaluation constraint. At any equilibrium $\theta^*$, we have:

$$J(\pi_{\theta^*}) - \beta\zeta^2_{\pi_{\theta^*},\psi_{\phi^*}} \ge J(\pi^*) - \beta\zeta^2_{\pi^*,\psi_{\phi^*}}. \tag{41}$$

Rearranging:

$$J(\pi_{\theta^*}) \ge J(\pi^*) - \beta\left(\zeta^2_{\pi^*,\psi_{\phi^*}} - \zeta^2_{\pi_{\theta^*},\psi_{\phi^*}}\right). \tag{42}$$

From Step 4, as $T \to \infty$, we have $\zeta^2_{\pi_{\theta^*},\psi_{\phi^*}} \to 0$. The key insight of co-learning is that the predictor continuously adapts to the evolving policy. As the policy $\pi_{\theta^*}$ approaches the optimal policy $\pi^*$ through the optimization process, the predictor $\phi^*$ learns to accurately evaluate policies in this region of policy space. Specifically, because the replay buffer $\mathcal{B}_\psi$ contains data from recent policies including those near $\pi^*$, the predictor also learns to evaluate $\pi^*$ accurately, hence $\zeta^2_{\pi^*,\psi_{\phi^*}} \to 0$ as $T \to \infty$.

Therefore, the penalty term vanishes with training iterations:

$$J(\pi_{\theta^*}) \ge J(\pi^*) - \epsilon_{\text{perf}}, \tag{43}$$

where $\epsilon_{\text{perf}} = \beta\left(\zeta^2_{\pi^*,\psi_{\phi^*}} - \zeta^2_{\pi_{\theta^*},\psi_{\phi^*}}\right) \to 0$ as $T \to \infty$, independent of the value of $\beta$.

This completes the proof. $\qquad\square$

**Remark.** This result formalizes the intuition that co-learning enables the predictor to "track" the evolving policy, maintaining low evaluation error, while the policy can explore the space of high-performing solutions without being constrained by a fixed predictor's limitations. The Stackelberg game framework provides a principled way to analyze this co-evolution and guarantees that any equilibrium of this process satisfies the desired properties of near-optimal performance and near-perfect evaluation.

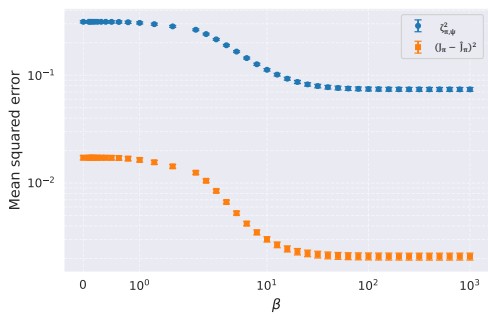

Figure 7: **Policy evaluation using predictor's value estimates.** Here, for the same experimental set-up described in the figure 6, we plot $\zeta_{\pi,\psi}^2$ and $(J_\pi - \hat{J}_\pi)^2$ as $\beta$ is varied. Each data point represents average over 1000 trials and the error bars denote the standard error. We find that as $\beta$ increases, the squared error between $\hat{J}_\pi$ w.r.t. $J_\pi$ reduces.

# B  JAX IMPLEMENTATION OF PREDICTABILITY TRANSFORMER

```python
import jax.numpy as jnp
import flax.linen as nn

class PredictabilityHead(nn.Module):

    num_heads:  int
    hidden_dim: int
    num_layers: int

    @nn.compact
    def __call__(
        self,
        assess_env_start_states: jnp.ndarray,   # [B,k,obs]
        assess_env_returns: jnp.ndarray,    # [B,k]
        query_state:    jnp.ndarray    # [B,obs]
    ) -> jnp.ndarray:              # [B]

        batch_size, k, obs_dim = assess_env_start_states.shape
        h = self.hidden_dim

        # ------------------------------------------------------
        # Token embeddings
        # ------------------------------------------------------
        assess_state_tokens  = nn.Dense(h)(assess_env_start_states)  # [B,k,h]
        assess_return_tokens = nn.Dense(h)(assess_env_returns[..., None])  #[B,k,h]
        query_state_token = nn.Dense(h)(query_state)[:, None, :]  # [B,1,h]

        # ------------------------------------------------------
        # Positional embeddings
        # ------------------------------------------------------
        pos_embeddings = nn.Embed(num_embeddings=k + 1,
                        features=h)(jnp.arange(k + 1))  # [k + 1,h]

        assess_state_tokens = assess_state_tokens + pos_embeddings[:-1, :]  #[B,k,h]
        assess_return_tokens = assess_return_tokens + pos_embeddings[-1, :]  # [B,k,h]
        query_state_token = query_state_token + pos_embeddings[-1, None, :]  # [B,1,h]

        # ------------------------------------------------------
        # Full Sequence
        # ------------------------------------------------------
        full_sequence = jnp.concatenate([assess_state_tokens,
        assess_return_tokens, query_state_token], axis=1)    # [B,2k+1,h]

        # ------------------------------------------------------
        # Transformer
        # ------------------------------------------------------
        x = full_sequence
        for _ in range(self.num_layers):
            # Self-attention block
            y = nn.LayerNorm()(x)
            y = nn.SelfAttention(num_heads=self.num_heads,
                                qkv_features=h)(y)
            x = x + y
            # Feed-forward block
            y = nn.LayerNorm()(x)
            y = nn.relu(nn.Dense(h)(y))
            y = nn.Dense(h)(y)
            x = x + y

        query_representation = x[:, -1, :]   # last (query) token
        value_prediction     = nn.Dense(1)(query_representation)  # [B,1]
        return jnp.squeeze(value_prediction, -1)               # [B]
```

## C  EvA-RL Algorithm

---

**Algorithm 1** Policy optimization with EvA-RL

---

**Require:** Assessment start-states $\{s_A^1, \ldots, s_A^k\}$, policy $\pi_\theta$, predictor $\psi_\phi$
1: Initialize an empty buffer $\mathcal{B}_\psi$
2: **for** $t = 0, 1, 2, \ldots$ **do**
3:   **if** $D_\psi$ has sufficient data **then**
4:     # PREDICTOR UPDATE
5:     **for** $i = 1$ to $N_{\text{pred}}$ **do**
6:       Sample a batch from buffer $\mathcal{B}_\psi$: $(\{s_A^i\}_{i=1}^k, \{g(h_A^i)\}_{i=1}^k, s_D, g(h_D))$.
7:       Update predictor $\phi$ via gradient descent

$$\phi_{t+1} = \phi_t - \alpha_\phi \nabla_\phi \left[ \mathbb{E}_{(\{s_A^i\}_{i=1}^k, \{g(h_A^i)\}_{i=1}^k, s_D, g(h_D)) \sim D_\psi} \left( g(h_D) - \psi_{\phi_t}(s_D, \{s_A^i\}_{i=1}^k, \{g(h_A^i)\}_{i=1}^k) \right)^2 \right]_{\phi = \phi_t} \tag{44}$$

8:     **end for**
9:   **end if**
10:   # POLICY UPDATE
11:   **for** $j = 1$ to $N_{\text{policy}}$ **do**
12:     Collect on-policy rollouts from deployment environment and assessment environment using current policy $\pi_\theta$.
13:     **if** Predictor update took place **then**
14:       Update $\theta$ via gradient ascent

$$\theta_{t+1} = \theta_t + \alpha_\theta \nabla_\theta \left[ \mathbb{E}_{s_D \sim \mu_D; h_D \sim \pi_\theta | s_D, M_D} \left( g(h_D) - \beta \left( g(h_D) - \psi_{\phi_{t+1}}(s_D, \{s_A^i\}_{i=1}^k, \{g(h_A^i)\}_{i=1}^k) \right)^2 \right) \right]_{\theta = \theta_t}. \tag{45}$$

15:     **else**
16:       Update $\theta$ via gradient ascent using only the policy gradient (without adding the gradient of predictability loss)
17:     **end if**
18:     Append assessment rollouts $\{g(h_A^i)\}_{i=1}^k$, deployment states $s_D$ and their returns $g(h_D)$ to buffer $\mathcal{B}_\psi$
19:   **end for**
20: **end for**

---

## D  Hyperparameter Details

In this section, we provide details of the hyperparameters pertaining to the EvA-RL algorithm. The hyperparameters for policy gradient optimization using advantage-actor critic follow the default values in the PureJaxRL library (Lu et al., 2022).

## D.1 DISCRETE ACTION SPACE

| Sr. No. | Hyperparameter | Value |
|---|---|---|
| 1 | Number of Environments | 64 |
| 2 | Number of Steps per Environment | 100 |
| 3 | Total Timesteps | 1e7 |
| 4 | Assessment Trajectory Length | 10 |
| 5 | General Trajectory Length | 200 |
| 6 | Learning Rate | 1e-4 |
| 7 | Number of Epochs | 100 |
| 8 | Number of Heads in Transformer | 4 |
| 9 | Number of Layers in Transformer | 4 |
| 10 | Hidden Dimension in Transformer | 16 |
| 11 | Number of Assessment Start States | 5 |
| 12 | Batch Size | 256 |
| 13 | Predictability Coefficient | 0, 1e-2, 1e-1 |
| 14 | Number of Predictor Updates | 5 |
| 15 | Seed | 0, 1, 2, 3, 4 |

## D.2 CONTINUOUS ACTION SPACE

| Sr. No. | Hyperparameter | Value |
|---|---|---|
| 1 | Number of Environments | 2048 |
| 2 | Number of Steps per Environment | 10 |
| 3 | Total Timesteps | 2e7 |
| 4 | Assessment Trajectory Length | 25 |
| 5 | General Trajectory Length | 1000 |
| 6 | Batch Size | 256 |
| 7 | Learning Rate | 1e-4 |
| 8 | Number of Epochs | 100 |
| 9 | Number of Heads in Transformer | 4 |
| 10 | Number of Layers in Transformer | 4 |
| 11 | Hidden Dimension in Transformer | 16 |
| 12 | Number of Assessment Start States | 5 |
| 13 | Batch Size | 256 |
| 14 | Predictability Coefficient | 0, 5e-4 |
| 15 | Seed | 0, 1, 2, 3, 4 |

# E    EXAMPLES OF ASSESSMENT ENVIRONMENT STATES

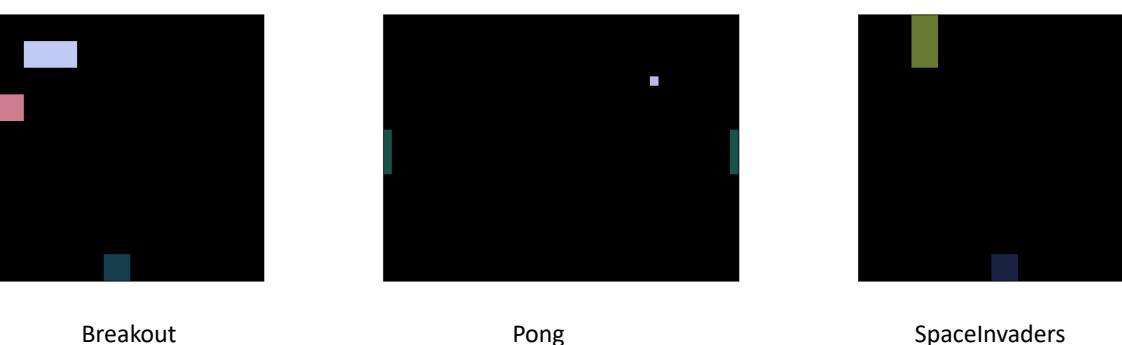

Figure 8: Representative assessment environment start-states for Gymnax environments of Breakout, Pong and SpaceInvaders. In the Breakout environment, we have a ball (in pink) bouncing off the bricks (in white) and the paddle (in grey). In the Pong environment, we have a ball (in white) and two paddles (in grey). In the SpaceInvaders environment, we have an alien spaceship (in green) and the player (in grey). We can use these states to test if the agent can break the bricks in the upper left corner in Breakout, or if the agent can handle ball coming from an edge in Pong, or if the agent can fire at an emancipated alien ship in SpaceInvaders. We will use these behaviors to form a behavioral encoding which we will use to estimate the agent's performance in true deployment environment.

## F    ADDITIONAL EXPERIMENTAL RESULTS

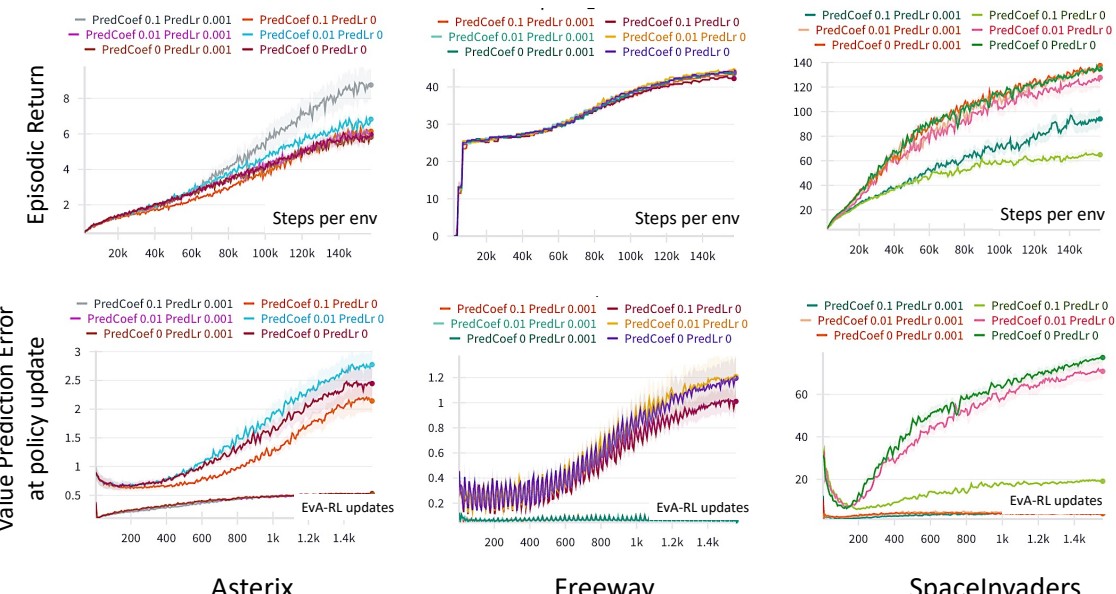

Figure 9: Learning curves for discrete action spaces. The plot shows the episodic return and prediction error computed immediately before updating a policy. We show mean and standard error across 5 different seeds for Asterix, Freeway and SpaceInvaders. The plots show the learning curves for 10 million interactions with the environment. Each plot has predictability coefficient varying among 0, 0.01 and 0.1 and the value predictor learning rate is 0 or 1e-3. In case of learning rate 0, the predictor is pretrained on data from standard policy gradient RL. Moreover, predictability coefficient 0 corresponds to the standard policy gradient RL. As described in section 4.1, we find that returns of the policies, where the predictor is updated along with the policy closely follow the returns of standard RL policies, while their prediction errors are significantly lower. Increase in predictability coefficient leads to decrease in the return while minimal gains are observed in lowering of the prediction error. Also, when we use a pretrained predictor which is kept frozen during training, the returns are lower than that of simultaneously updated predictor. More importantly, the predictor is being trained on past policies and the error is close to zero when the predictor is co-learned. This allows us to reliably provide evaluation feedback to the policy.

## G    SAFETY APPLICATION: DEPLOYMENT GATING WITH A SEPARATE ASSESSMENT ENVIRONMENT

We illustrate EvA-RL's usefulness for safety via a deployment-gating example where the assessment environment is a separate $2 \times 2$ grid, distinct from the deployment MDP. Assessment episodes are length 1, and assessment states exist only in the assessment environment.

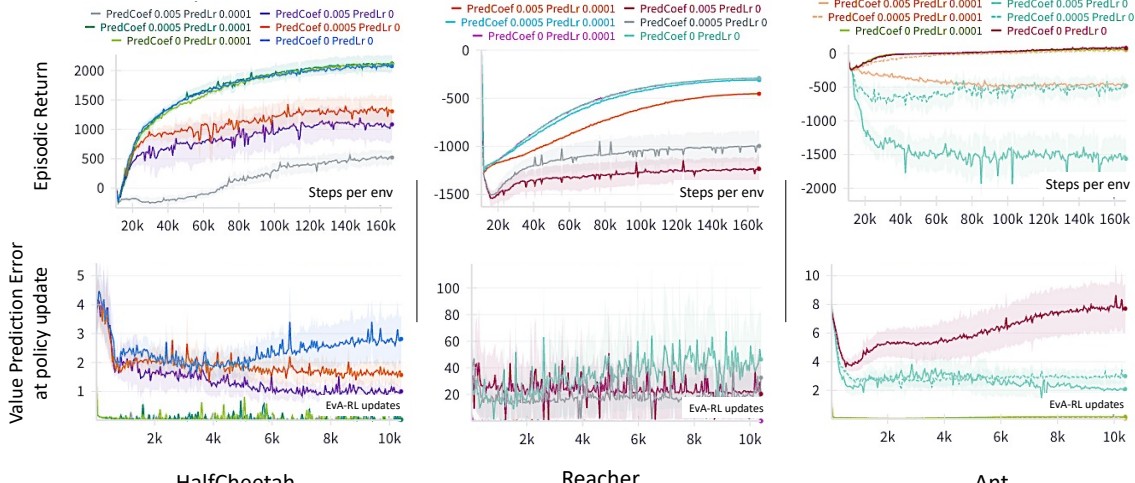

Figure 10: Training curves for continuous action space environments. The plot shows mean and standard error of the episodic return and prediction error computed immediately before updating a policy across 5 different seeds for the Brax environments of halfcheetah, reacher and ant. The plots show the learning curves for 10 million interactions with the environment. Each plot has predictability coefficient varying among 0, 0.0005 and 0.005, and the value predictor learning rate is 0 or 1e-4. For the case of learning rate to be 0, the predictor is pretrained on data from standard policy gradient RL. The curves here follow similar trends as the discrete action space learning curves described in figure 9. In addition, we find that these environments are more sensitive to changes in the predictability coefficient. The learning rate for the predictor is also lowered compared to the discrete case due to different order of magnitude of the rewards.

## G.1 A CASE STUDY ON DEPLOYMENT GATING USING EvA-RL

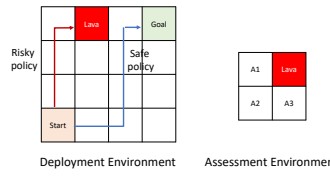

Deployment Environment          Assessment Environment

Figure 11: The deployment environment is a 4x4 gridworld with a start state $S = (3, 0)$, goal state $G = (0, 3)$ with reward $+1$ and a lava state $L = (0, 1)$ with reward $-10$. The assessment environment is a 2x2 gridworld with assessment states at three corners, lava at one corner and episode length 1. We train a linear value predictor on the assessment behavior of 3 deployable, safe policies. Then, we consider two candidate policies for deployment – one safe and one risky – and demonstrate that the trained predictor's estimates can be used to avoid deploying the risky policy without performing any additional rollouts in the deployment environment.

**Deployment environment.** We reuse the $4 \times 4$ gridworld:

- States: $(i, j)$, $i, j \in \{0, 1, 2, 3\}$.
- Start: $S = (3, 0)$, goal: $G = (0, 3)$ with reward $+1$.
- Single lava state: $L = (0, 1)$ (on the top border, near $G$) with reward $-10$.
- All other transitions give reward 0; episodes terminate at $G$ or $L$.

- Discount factor: $\gamma = 1$.

We define five deterministic policies: three *training-safe* policies, one *evaluation-safe* policy, and one *risky* policy. Their trajectories from $S$ in the deployment environment are:

- **Safe-train 1**:
$$(3,0) \to (2,0) \to (1,0) \to (1,1) \to (1,2) \to (1,3) \to (0,3) = G$$

- **Safe-train 2**:
$$(3,0) \to (3,1) \to (3,2) \to (2,2) \to (1,2) \to (0,2) \to (0,3) = G$$

- **Safe-train 3**:
$$(3,0) \to (2,0) \to (2,1) \to (2,2) \to (1,2) \to (0,2) \to (0,3) = G$$

- **Safe-eval**:
$$(3,0) \to (3,1) \to (2,1) \to (1,1) \to (1,2) \to (0,2) \to (0,3) = G$$

- **Risky**:
$$(3,0) \to (2,0) \to (1,0) \to (0,0) \to (0,1) = L$$

Thus

$$J^{\text{safe-train},k}(S) = +1, \quad k = 1,2,3, \tag{46}$$

$$J^{\text{safe-eval}}(S) = +1, \qquad J^{\text{risky}}(S) = -10. \tag{47}$$

**Assessment environment.** The assessment environment is a separate $2 \times 2$ grid:

- States: $(i,j)$, $i,j \in \{0,1\}$.
- Lava: $\tilde{L} = (0,1)$ with reward $-10$ (terminal).
- Assessment states: the three remaining corners

$$\tilde{A}_1 = (0,0), \quad \tilde{A}_2 = (1,0), \quad \tilde{A}_3 = (1,1).$$

- Actions: up/down/left/right with standard grid transitions (clipped at borders).
- Episode length: 1 step. Reward is $+1$ if the next state is not $\tilde{L}$, and $-10$ if the next state is $\tilde{L}$. Episodes terminate after this single step.

Each policy $\pi$ induces a one-step "path" from each assessment state:

$$\tilde{A}_j \xrightarrow{\pi} \text{next state}, \quad j \in \{1,2,3\}.$$

We define the assessment behavior as follows.

**Assessment paths and returns.** For the three training-safe policies:

- **Safe-train 1:**
$$\tilde{A}_1 = (0,0) \to (1,0), \quad \tilde{A}_2 = (1,0) \to (1,1), \quad \tilde{A}_3 = (1,1) \to (1,0)$$

- **Safe-train 2:**
$$\tilde{A}_1 = (0,0) \to (1,0), \quad \tilde{A}_2 = (1,0) \to (1,1), \quad \tilde{A}_3 = (1,1) \to (0,0)$$

- **Safe-train 3:**
$$\tilde{A}_1 = (0,0) \to (1,0), \quad \tilde{A}_2 = (1,0) \to (0,0), \quad \tilde{A}_3 = (1,1) \to (1,0)$$

All three training-safe policies avoid $\tilde{L}$ from every assessment state, so
$$(R^{\text{safe-train},k}_{\tilde{A}_1}, R^{\text{safe-train},k}_{\tilde{A}_2}, R^{\text{safe-train},k}_{\tilde{A}_3}) = (+1,+1,+1), \quad k = 1,2,3. \tag{48}$$

For the evaluation-safe policy:

- **Safe-eval:**
$$\tilde{A}_1 = (0,0) \to (1,0), \quad \tilde{A}_2 = (1,0) \to (1,1), \quad \tilde{A}_3 = (1,1) \to (0,0),$$

it also avoids $\tilde{L}$ everywhere, so
$$(R^{\text{safe-eval}}_{\tilde{A}_1}, R^{\text{safe-eval}}_{\tilde{A}_2}, R^{\text{safe-eval}}_{\tilde{A}_3}) = (+1,+1,+1). \tag{49}$$

For the risky policy we let it be catastrophically wrong from $\tilde{A}_1$:

- **Risky:**
$$\tilde{A}_1 = (0,0) \to (0,1) = \tilde{L}, \quad \tilde{A}_2 = (1,0) \to (1,1), \quad \tilde{A}_3 = (1,1) \to (1,0),$$

so
$$(R^{\text{risky}}_{\tilde{A}_1}, R^{\text{risky}}_{\tilde{A}_2}, R^{\text{risky}}_{\tilde{A}_3}) = (-10,+1,+1). \tag{50}$$

**Predictor training.** We fit a linear predictor from assessment returns to deployment return:
$$\hat{J}^\pi = b + w_1 R^\pi_{\tilde{A}_1} + w_2 R^\pi_{\tilde{A}_2} + w_3 R^\pi_{\tilde{A}_3}, \tag{51}$$
using only the three training-safe policies, for which
$$(R_{\tilde{A}_1}, R_{\tilde{A}_2}, R_{\tilde{A}_3}) = (+1,+1,+1), \quad J(S) = +1.$$
Thus any $(b, w_1, w_2, w_3)$ satisfying
$$1 = b + w_1 + w_2 + w_3 \tag{52}$$
fits the training data. A natural minimum-norm symmetric choice is
$$b = 0, \qquad w_1 = w_2 = w_3 = \tfrac{1}{3}, \tag{53}$$
yielding
$$\hat{J}^\pi = \frac{R^\pi_{\tilde{A}_1} + R^\pi_{\tilde{A}_2} + R^\pi_{\tilde{A}_3}}{3}. \tag{54}$$

**Evaluation and deployment gating.** For the evaluation-safe policy,
$$(R^{\text{safe-eval}}_{\tilde{A}_1}, R^{\text{safe-eval}}_{\tilde{A}_2}, R^{\text{safe-eval}}_{\tilde{A}_3}) = (+1,+1,+1) \implies \hat{J}^{\text{safe-eval}} = 1, \tag{55}$$
matching $J^{\text{safe-eval}}(S) = +1$.

For the risky policy,
$$(R^{\text{risky}}_{\tilde{A}_1}, R^{\text{risky}}_{\tilde{A}_2}, R^{\text{risky}}_{\tilde{A}_3}) = (-10,+1,+1) \implies \hat{J}^{\text{risky}} = \frac{-10+1+1}{3} = -\frac{8}{3} \approx -2.67, \tag{56}$$
while $J^{\text{risky}}(S) = -10$. Although the predictor underestimates the severity of the catastrophe, it still assigns a strictly negative value, clearly separated from the safe policies' value of $+1$.

**Gating rule.** An evaluation-aware deployment rule
$$\text{Deploy policy } \pi \text{ only if } \hat{J}^\pi > 0 \tag{57}$$

- approves the evaluation-safe policy ($\hat{J}^{\text{safe-eval}} = 1 > 0$), and

- rejects the risky policy ($\hat{J}^{\text{risky}} = -\frac{8}{3} < 0$).

This example shows how a separate, short-horizon assessment environment—here a $2 \times 2$ grid with a single lava state and three assessment states—combined with a simple predictor trained only on safe policies, can gate deployment and systematically exclude policies whose assessment behavior signals high risk.

## H    DISCLOSURE ABOUT LLM USAGE

In this work, we used LLMs to develop the code-base for the experiments, check the correctness of the theoretical proofs and to polish the manuscript.

